# GraphDE: A Generative Framework for Debiased Learning and Out-of-Distribution Detection on Graphs

**Zenan Li, Qitian Wu, Fan Nie, Junchi Yan**[*]

Department of Computer Science and Engineering and MoE Key Lab of Artificial Intelligence
Shanghai Jiao Tong University
{emiyali, echo740, youluo2001, yanjunchi}@sjtu.edu.cn

## Abstract

Despite the remarkable success of graph neural networks (GNNs) for graph representation learning, they are generally built on the (unreliable) i.i.d. assumption across training and testing data. However, real-world graph data are universally comprised of outliers in training set and out-of-distribution (OOD) testing samples from unseen domains, which solicits effective models for i) debiased learning and ii) OOD detection, towards general trustworthy purpose. In this paper, we first mathematically formulate the two challenging problems for graph data and take an initiative on tackling them under a unified probabilistic model. Specifically, we model the graph generative process to characterize the distribution shifts of graph data together with an additionally introduced latent environment variable as an indicator. We then define a variational distribution, i.e., a recognition model, to infer the environment during training of GNN. By instantiating the generative models as two-component mixtures, we derive a tractable learning objective and theoretically justify that the model can i) automatically identify and down-weight outliers in the training procedure, and ii) induce an effective OOD detector simultaneously. Experiments on diverse datasets with different types of OOD data prove that our model consistently outperforms strong baselines for both debiasing and OOD detection tasks. The source code has been made publicly available at https://github.com/Emiyalzn/GraphDE.

## 1 Introduction

For processing ubiquitous graph data, graph neural networks (GNNs) [17, 43] have emerged as effective approaches for graph representation learning [12], which in turn has spanned a wide range of applications, from drug discovery [23] to image classification [4].

Despite their success, existing GNNs are generally built on the assumption that both training and testing data are independently sampled from the identical distribution (i.i.d.), which often does not hold [45, 22, 49, 48]. Real-world training data is often shown universally mixed with outliers [30, 7], which specifically may incur large gradient steps in the wrong direction [38], or be overfitted by the model [3], thereby hurting models' performance. Besides, testing samples from unseen domains are also unavoidable when the model is deployed in the wild [2, 19], which may easily be classified into a wrong class with a high probability [13]. Inevitably, graph data also involve these two common problems. As illustrated in Fig. 1, we indeed find that vanilla GNNs perform poorly when encountering large portion of outliers and out-of-distribution (OOD) testing samples, which solicits effective models for i) debiased learning and ii) OOD detection, towards general trustworthy purpose.

---

[*]Junchi Yan is the corresponding author who is also with Shanghai AI Laboratory.

36th Conference on Neural Information Processing Systems (NeurIPS 2022).

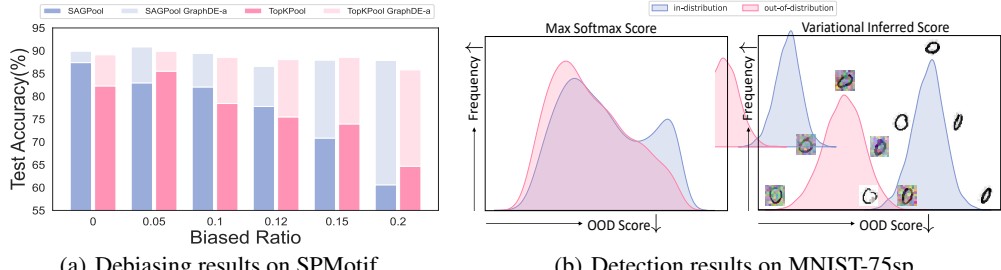

(a) Debiasing results on SPMotif  (b) Detection results on MNIST-75sp

Figure 1: Vanilla GNNs' performance degenerates drastically with a larger number of outliers in the training dataset. They also assign high softmax scores to OOD data, which may raise trustworthy concerns. In comparison, our proposed framework can boost performance for both tasks notably. In (a), GraphDE-a is plugged in the backbone; In (b), the variational inferred score is derived from GraphDE's recognition model. We plot MNIST images at corresponding OOD scores for intuition.

However, these two problems still remain less explored for graph data. Generally, identifying graph outliers (or OOD samples) is inherently hard as the non-Euclidean graph data can own a large number of node features and additional structural knowledge. The bias may exist in certain node features, local and global graph structures [29], as well as the underlying relations between features, structures, and labels [38], preventing trivial applications of traditional machine learning methods [25, 6] from identifying outliers on graph data.

We argue that training outliers and OOD testing samples can be conceptually unified as they both represent data drawn from distributions other than the major in-distribution (ID) part of training data. However, they are inherently different as the training outliers also catch label-related distribution shifts, e.g. noisy labels [1], while OOD testing samples are solely determined by their features [14] since we cannot access their ground-truth labels. This difference also explains why many existing works tend to tackle the two problems independently.

In light of the intrinsic connection between these two types of data, we present GraphDE, a probabilistic generative framework for **Graph DE**biased learning and OOD **DE**tection. Specifically, we model the generative process to characterize the distribution shifts of graph data, along with an environment variable as an indicator. Utilizing variational inference [51], we propose to infer the environment variable during the training procedure of GNN. Our framework consists of three modules: the recognition model that infers the environment variable based on the input graph data; the structure estimation model by which we extract an OOD detector on testing dataset; the classification model which is our target GNN. Following the idea of data resampling [38, 37], we model the latter two modules by two-component mixtures, with one component for ID data, and the other component robustifying the model by isolating outliers. Based on this, we derive a tractable learning objective for our model and theoretically justify its effectiveness for the target tasks. **The contributions are:**

• We focus on two critical but largely unexplored problems for graph data: debiased learning and OOD detection, find their fundamental correlations and define them under a probabilistic framework.

• We propose GraphDE with a novel learning objective and theoretically justify that: 1) it can identify and down-weight outliers during training; 2) it can provide an OOD detector on the testing set; 3) its constituent modules are optimized in a mutually-promoting manner.

• We evaluate GraphDE with different GNN backbones on diverse datasets of different OOD types, where GraphDE achieves consistent performance improvements over the baselines. Specifically, as shown in Fig. 1, for the debiasing task, it yields up to 14.6% accuracy improvement on SPMotif; for the OOD detection task, it outperforms the strongest baseline by 9.31% on MNIST-75sp.

## 2 Problem Formulation and Related Works

In typical graph learning problems, given a set of labeled graphs $\mathcal{D}^{tr} = \{(G_i, y_i)\}_{i=1}^{N^{tr}}$, we need to train a model $f_\theta : G \to \hat{y}$ to predict the labels on the testing dataset $\mathcal{D}^{te} = \{G_i\}_{i=1}^{N^{te}}$. Specifically, the input graph $G = (A, X)$ is composed of an adjacency matrix $A = \{a_{vu} | v, u \in V\}$ and node features

$X = \{x_v | v \in V\}$, where $V$ is the node set. Define $\mathbf{G}$ as a random variable of the input graph and $\mathbf{y}$ as a random variable of the graph label, the data-generating process can then be characterized as $p(\mathbf{G}, \mathbf{y}|\mathbf{e}) = p(\mathbf{G}|\mathbf{e})p(\mathbf{y}|\mathbf{G}, \mathbf{e})$ where we introduce an environment variable $\mathbf{e}$ as a potential unknown indicator for data from particular distributions.

The biased data (we use the form interchangeable to outliers) in the training set and out-of-distribution (OOD) data in the testing set can both be treated as data drawn from different distributions to the distribution generating the majority of the training set and testing data of our interests, i.e., the in-distribution (ID) data so-called. Hence, we can model them by a binary environment variable $\mathbf{e}$, with $\mathbf{e} = 1$ denoting the ID portion and $\mathbf{e} = 0$ for the biased/OOD portion. Then the training and testing datasets can be split into $\mathcal{D}^{tr} = \mathcal{D}_{in}^{tr} \cup \mathcal{D}_{out}^{tr}$ and $\mathcal{D}^{te} = \mathcal{D}_{in}^{te} \cup \mathcal{D}_{out}^{te}$ accordingly. Note that i) the partitions of training/testing data are only for illustration purpose and are strictly unknown in practice; ii) the testing data and distribution are also strictly unknown during the training process; iii) here $\mathbf{e}$ is mainly used for distinguishing the ID part with the OOD counterpart of datasets. In fact, the OOD part ($\mathbf{e} = 0$) may consist of data from multiple distributions [2].

Denote the loss on sample $(G, y)$ by $L(y, f_\theta(G))$, e.g., cross-entropy loss for classification problems. Then the standard empirical risk minimization (ERM) objective can be written as

$$\min_\theta \mathbb{E}_{e \sim p(\mathbf{e})} \Big[ \mathbb{E}_{(G,y) \sim p(\mathbf{G}, \mathbf{y}|\mathbf{e}=e)} [L(y, f_\theta(G))] \Big]. \tag{1}$$

This objective works based on the i.i.d. assumption across training and testing data. As we aim at promoting the ID test accuracy (i.e. the performance on $\mathcal{D}_{in}^{te}$), however, the updating gradient by training data drawn from $p(\mathbf{G}, \mathbf{y}|\mathbf{e} = 0)$ will hurt model performance drastically [3]. On the other hand, for testing data from unseen distributions $\mathcal{D}_{out}^{te}$, the model would incline to produce imprecise results as $p^{tr}(\mathbf{e}) \neq p_{out}^{te}(\mathbf{e})$, which is undesired and should be avoided for the trustworthy purpose. We next formulate the two problems of our interests.

**Task I: Debiased learning for training data.** The target is to mitigate outliers' negative effects during training, thereby enhancing the performance on ID testing data. We modify the ERM objective:

$$\min_\theta \mathbb{E}_{e \sim p(\mathbf{e})} \Big[ \mathbb{E}_{(G,y) \sim p(\mathbf{G}, \mathbf{y}|\mathbf{e}=e)} [L(y, f_\theta(G)|e)] \Big],$$
$$\text{where } L(y, f_\theta(G)|e) = \begin{cases} L(y, f_\theta(G)), & \text{if } e = 1, \\ L_0, & \text{if } e = 0. \end{cases} \tag{2}$$

where the loss function for biased training data is fixed as $L_0$, so the debiasing objective is equivalent to minimize $\mathbb{E}_{(G,y) \sim p(\mathbf{G}, \mathbf{y}|\mathbf{e}=1)} [L(y, f_\theta(G))]$, i.e., the training loss for the ID portion $\mathcal{D}_{in}^{tr}$.

**Task II: OOD detection for testing data.** We aim to obtain an OOD detector $g$ composed of a trained model $f_\theta : G \to \hat{y}$, a scoring function $s$ and a threshold $\tau$:

$$g(G; \tau, s, f_\theta) = \begin{cases} 0 \text{ (OOD)}, & \text{if } s(G; f_\theta) \leq \tau, \\ 1 \text{ (ID)}, & \text{if } s(G; f_\theta) > \tau. \end{cases} \tag{3}$$

As we are willing to induce an OOD detector during the training procedure of GraphDE, we leverage the trained model $f_\theta$ from task I and what we need is to find the optimal score function $s^*$ that can best separate out ID and OOD data (e.g. maximize the AUROC).

**Related works on debiased learning.** A series of strategies have been proposed to debias the training dataset in general machine learning [38, 5] and computer vision [30, 7]. Many works propose to preprocess the dataset by filtering out the outliers [5, 25]. However, these methods cannot capture label-related distribution variations such as noisy labels [1] and also cannot be easily adapted to deal with graph data. Another popular strategy is data resampling [38, 37], which relies on the training loss to identify and down-weight outliers in the training set. However, these methods only capture bias in the mapping from features to labels, but ignore the variation in feature space. For the first time to our best knowledge, we formally define and deal with the graph debiased learning problem.

**Related works on OOD detection.** A trustworthy learning system should not only produce accurate predictions for ID data, but also distinguish and reject OOD examples without further prediction.

---

[2]Since we are not interested in the specific OOD distributions, we model them by a unified distribution $p(\mathbf{G}, \mathbf{y}|\mathbf{e} = 0)$, which is enough to identify the OOD samples as we will justify later.

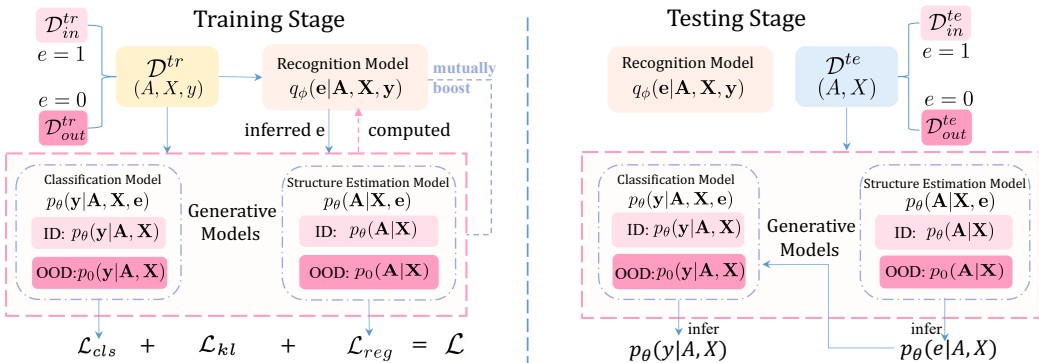

Figure 2: The proposed GraphDE framework. **Left:** During the training stage, the training data is fed into the recognition model to infer the environment variable, then fed into the generative models to compute the total loss for training (we omit the source of $\mathcal{L}_{kl}$ in the figure for simplicity). **Right:** In the testing stage, the testing data is fed into the structure estimation model for OOD detection, then fed into the classification model to compute the class probabilities.

Deep learning based OOD detection has achieved promising progress in vision [27, 13]. Besides, node and edge-level anomaly detection on graphs are also widely studied, some of which are based on classical network science [26], while others utilize the power of GNNs [36, 9]. In comparison, only limited works [52, 29] attempt to solve graph-level OOD detection. Specifically, OCGIN [52] designs a one-class GNN objective to distinguish between ID and OOD data. GLocalKD [29] utilizes joint random distillation of graph and node representations to learn rich global and local pattern information of graph-structured data. However, both of them need to be trained individually for the detection task. In this paper, we extract an OOD detector during the training procedure of the target GNN, while outperforming all the baselines on 7 out of 9 benchmarks. Appendix B gives more comprehensive description on related works on GNNs and Trustworthy GNNs.

## 3 Methodology

### 3.1 Probabilistic Generative Learning for Debiased Learning and OOD Detection

To improve the ID test accuracy, we want to make the model more robust by automatically identifying potential outliers during training. Thus, we take a generative perspective on the graph data and adopt variational inference to infer the environment variable during training.

**Generative Process.** We model the joint distribution $p(\mathbf{X}, \mathbf{y}, \mathbf{A}, \mathbf{e})$ to capture graph-level distribution shifts. In line with the homophily principle from random graph theory [33], i.e., linking probability depends on some inherent properties of nodes, we assume that the graph structure is generated based on the node features and the environment variable. The graph labels are then generated based on the graph structure, node features, and the environment. We also assume $p(\mathbf{e})$ and $p(\mathbf{X})$ to be independent distributions for environments and node features, respectively. We do not specify the form of $p(\mathbf{X})$, since everything will be conditioned on $\mathbf{X}$ in this paper, similar to [28, 44]. We have:

$$p(\mathbf{X}, \mathbf{y}, \mathbf{A}, \mathbf{e}) = p(\mathbf{e})p(\mathbf{X})p(\mathbf{A}|\mathbf{X}, \mathbf{e})p(\mathbf{y}|\mathbf{X}, \mathbf{A}, \mathbf{e}), \quad (4)$$

where the generative models $p(\mathbf{A}|\mathbf{X}, \mathbf{e})$ and $p(\mathbf{y}|\mathbf{X}, \mathbf{A}, \mathbf{e})$ could be modeled by some flexible parametric distributions $p_\theta(\mathbf{A}|\mathbf{X}, \mathbf{e})$ and $p_\theta(\mathbf{y}|\mathbf{X}, \mathbf{A}, \mathbf{e})$ with parameter $\theta$. These two distributions will later be instantiated as the structure estimation model and classification model, respectively. We define the generative models with two-component mixtures. For example, $p_\theta(\mathbf{A}|\mathbf{X}, \mathbf{e})$ is composed of an ID component $p_\theta(\mathbf{A}|\mathbf{X})$, and an OOD component $p_0(\mathbf{A}|\mathbf{X})$ (the subscript 0 implies a fixed distribution). The environment variable $\mathbf{e} \in \{0, 1\}$ indicates whether the graph should be modeled by the ID ($\mathbf{e} = 1$) or the OOD ($\mathbf{e} = 0$) component. Specifically, these two terms are decomposed as:

$$p_\theta(\mathbf{A}|\mathbf{X}, \mathbf{e}) = p_\theta(\mathbf{A}|\mathbf{X})^{\mathbf{e}} p_0(\mathbf{A}|\mathbf{X})^{1-\mathbf{e}},$$
$$p_\theta(\mathbf{y}|\mathbf{X}, \mathbf{A}, \mathbf{e}) = p_\theta(\mathbf{y}|\mathbf{X}, \mathbf{A})^{\mathbf{e}} p_0(\mathbf{y}|\mathbf{X}, \mathbf{A})^{1-\mathbf{e}}. \quad (5)$$

**Recognition Model.** As the variables $\mathbf{X}$, $\mathbf{A}$ and $\mathbf{y}$ are all observed during training, what we need is to infer the sample-wise environment variable $\mathbf{e}$ conditional on these observed variables. Thus, we resort to variational inference [51] and introduce a recognition model $q_\phi(\mathbf{e}|\mathbf{A}, \mathbf{X}, \mathbf{y})$ parameterized with $\phi$ to approximate the true posterior $p_\theta(\mathbf{e}|\mathbf{A}, \mathbf{X}, \mathbf{y})$.

**Learning Objective.** We train the model parameters $\theta$ and $\phi$ by optimizing the Evidence Lower BOund (ELBO) of the observed data tuple $(A, X, y)$:

$$
\begin{aligned}
\log p_\theta(A, y|X) &\geq \log p_\theta(A, y|X) - D_{\mathrm{KL}}[q_\phi(\mathbf{e}|A, X, y)||p_\theta(\mathbf{e}|A, X, y)] \\
&= \mathbb{E}_{q_\phi(\mathbf{e}|A,X,y)}[\log(p_\theta(A|X, \mathbf{e})p_\theta(y|X, A, \mathbf{e}))] \\
&\quad - D_{\mathrm{KL}}[q_\phi(\mathbf{e}|A, X, y)||p(\mathbf{e})] = \mathcal{L}_{\mathrm{ELBO}}.
\end{aligned}
\tag{6}
$$

Eq. 6 shows that by maximizing the ELBO, we are maximizing the log-likelihood of observed data $\log p_\theta(A, y|X)$ while minimizing the KL divergence between the recognition model and the true posterior $p_\theta(\mathbf{e}|A, X, y)$. Finally, the learning objective for the training dataset can be written as:

$$
\begin{aligned}
-\mathcal{L} = &\frac{1}{N^{tr}} \sum_{i=1}^{N^{tr}} \underbrace{\mathbb{E}_{q_\phi(\mathbf{e}_i|A_i, X_i, y_i)}\left[\mathbf{e}_i \log p_\theta(y_i|X_i, A_i) + (1 - \mathbf{e}_i) \log p_0(y_i|X_i, A_i)\right]}_{\text{Classification Loss } \mathcal{L}_{cls}} \\
&+ \frac{1}{N^{tr}} \sum_{i=1}^{N^{tr}} \underbrace{\mathbb{E}_{q_\phi(\mathbf{e}_i|A_i, X_i, y_i)}\left[\mathbf{e}_i \log p_\theta(A_i|X_i) + (1 - \mathbf{e}_i) \log p_0(A_i|X_i)\right]}_{\text{Structure Regularization Loss } \mathcal{L}_{reg}} \\
&- \frac{1}{N^{tr}} \sum_{i=1}^{N^{tr}} \underbrace{D_{\mathrm{KL}}\left[q_\phi(\mathbf{e}_i|A_i, X_i, y_i)||p(\mathbf{e})\right]}_{\text{KL Loss } \mathcal{L}_{kl}}.
\end{aligned}
\tag{7}
$$

**Justification of GraphDE.** We provide theoretical analysis here to reveal the rationale of GraphDE. Without loss of generality, we do not take account into the KL loss here for simplicity. The proofs of the propositions are postponed to Appendix A.

**Proposition 1.** *1) The learning objective for GraphDE is in a re-weighted form when $q_\phi(\mathbf{e}|\mathbf{A}, \mathbf{X}, \mathbf{y})$ is instantiated as a Bernoulli distribution, with $q_\phi(\mathbf{e}_i = 1|A_i, X_i, y_i)$ acting as a weight for the $i$-th sample; 2) Given the ideal recognition model $q_\phi^*$ that gives $q_\phi^*(\mathbf{e} = 1|(A, X, y) \in \mathcal{D}_{in}) = 1$ and $q_\phi^*(\mathbf{e} = 1|(A, X, y) \in \mathcal{D}_{out}) = 0$, the generative models can learn to best fit the ID data.*

Prop. 1 reveals how the inferred posterior takes effect on the learning objective. Typically, if $q_\phi(\mathbf{e}|\mathbf{A}, \mathbf{X}, \mathbf{y})$ can assign a higher probability to ID data, our model can thereby learn to ignore the negative effects of biased training data and better fit the target distribution (i.e. ID).

**Proposition 2.** *1) Assuming the generative models fit to the ID data, i.e. $p_\theta(A|X \in \mathcal{D}_{in}) \geq p_\theta(A|X \in \mathcal{D}_{out})$ and $p_\theta(y|(A, X) \in \mathcal{D}_{in}) \geq p_\theta(y|(A, X) \in \mathcal{D}_{out})$, the recognition model will learn to predict $q_\phi(\mathbf{e} = 1|(A, X, y) \in \mathcal{D}_{in}) \geq q_\phi(\mathbf{e} = 1|(A, X, y) \in \mathcal{D}_{out})$; 2) Given optimal generative models that best fit the ID data and perform randomly on outliers, there exists a recognition model $q_\phi^*$ which yields the minimal objective while ideally predict the environment variable.*

Prop. 2 proves GraphDE's effectiveness from the recognition model's perspective. To summarize, the above two propositions show that $q_\phi(\mathbf{e}|\mathbf{A}, \mathbf{X}, \mathbf{y})$, $p_\theta(\mathbf{y}|\mathbf{A}, \mathbf{X})$ and $p_\theta(\mathbf{A}|\mathbf{X})$ can **mutually promote each other** during training. As deep networks tend to learn simple shared (i.e. ID) patterns first [3], this positive feedback can guide GraphDE to identify outliers in early epochs and thereby learn better representations of ID data. Besides, the propositions also suggest that given either of the recognition model or the generative model optimal, the other can be trained to be optimal, which suggests the existence of the global optimum. Based on these, we can further understand the implications of the three loss terms in Eq. 7: i) $\mathcal{L}_{cls}$ is the re-weighted version of the original classification loss; ii) $\mathcal{L}_{reg}$ acts as graph regularization and will impact the gradient of $q_\phi(\mathbf{e}|\mathbf{A}, \mathbf{X}, \mathbf{y})$, weighting ID data higher than the outliers; iii) the third term $\mathcal{L}_{kl}$ is the KL divergence between the learned posterior and the fixed prior distribution, ensuring the posterior not go too complex.

**Inference on Testing Dataset.** For new data, we need to infer the posterior conditional distribution of $\mathbf{e}$ and $\mathbf{y}$ given $(\mathbf{A}, \mathbf{X})$, i.e. $p_\theta(\mathbf{e}|\mathbf{A}, \mathbf{X})$ and $p_\theta(\mathbf{y}|\mathbf{A}, \mathbf{X})$, respectively. To compute $p_\theta(\mathbf{e}|\mathbf{A}, \mathbf{X})$ is

just to do **OOD detection**. For binary $\mathbf{e}$, we can compute its posterior analytically:

$$p_\theta(\mathbf{e}|\mathbf{A}, \mathbf{X}) = \frac{p_\theta(\mathbf{e}, \mathbf{A}, \mathbf{X})}{\sum_\mathbf{e} p_\theta(\mathbf{e}, \mathbf{A}, \mathbf{X})} = \frac{p(\mathbf{e})p(\mathbf{X})p_\theta(\mathbf{A}|\mathbf{X}, \mathbf{e})}{\sum_\mathbf{e} p(\mathbf{e})p(\mathbf{X})p_\theta(\mathbf{A}|\mathbf{X}, \mathbf{e})}. \tag{8}$$

The posterior can be used as the score function $s$ for the OOD detector, once we have the well-trained $p_\theta(\mathbf{A}|\mathbf{X}, \mathbf{e})$. Subsequently, we can calculate $p_\theta(\mathbf{y}|\mathbf{A}, \mathbf{X})$ by the total probability theorem:

$$p_\theta(\mathbf{y}|\mathbf{A}, \mathbf{X}) = \sum_\mathbf{e} p_\theta(\mathbf{y}|\mathbf{A}, \mathbf{X}, \mathbf{e})p_\theta(\mathbf{e}|\mathbf{A}, \mathbf{X}), \tag{9}$$

where $p_\theta(\mathbf{y}|\mathbf{A}, \mathbf{X}, \mathbf{e})$ can also be obtained after training.

### 3.2 GraphDE Instantiations

We proceed to specify the distributions $p_\theta(\mathbf{A}|\mathbf{X}, \mathbf{e})$, $p_\theta(\mathbf{y}|\mathbf{X}, \mathbf{A}, \mathbf{e})$, $p(\mathbf{e})$ and $q_\phi(\mathbf{e}|\mathbf{A}, \mathbf{X}, \mathbf{y})$. We assume $p(\mathbf{e}) = \text{Bernoulli}(\alpha)$ where $\alpha \in [0, 1]$ represents our prior belief about the cleanliness of the dataset. A higher $\alpha$ implies a larger portion of ID data in $\mathcal{D}^{tr}$. It is fixed as a scalar during training.

#### 3.2.1 Instantiations of the Recognition Model

We propose two variants for the recognition model $q_\phi(\mathbf{e}|\mathbf{A}, \mathbf{X}, \mathbf{y})$. In the first way, we directly assign a learnable scalar for each sample in the training dataset, i.e., $q_\phi(\mathbf{e}_i|A_i, X_i, y_i) = \text{Bernoulli}(\alpha_i)$, where $\alpha_i \in [0, 1]$ is a trainable parameter jointly optimized with generative models through the ELBO in Eq. 7. We call this variant GraphDE-v(ariational) as we optimize directly over $\alpha_i$. In the second way, notice that $\mathbf{e}$ is a binary variable by definition, we can compute the posterior analytically by

$$\begin{aligned} q_\phi(\mathbf{e}|\mathbf{A}, \mathbf{X}, \mathbf{y}) \doteq p_\theta(\mathbf{e}|\mathbf{A}, \mathbf{X}, \mathbf{y}) &= \frac{p_\theta(\mathbf{X}, \mathbf{y}, \mathbf{A}, \mathbf{e})}{\sum_\mathbf{e} p_\theta(\mathbf{X}, \mathbf{y}, \mathbf{A}, \mathbf{e})} \\ &= \frac{p(\mathbf{e})p(\mathbf{X})p_\theta(\mathbf{A}|\mathbf{X}, \mathbf{e})p_\theta(\mathbf{y}|\mathbf{X}, \mathbf{A}, \mathbf{e})}{\sum_\mathbf{e} p(\mathbf{e})p(\mathbf{X})p_\theta(\mathbf{A}|\mathbf{X}, \mathbf{e})p_\theta(\mathbf{y}|\mathbf{X}, \mathbf{A}, \mathbf{e})}. \end{aligned} \tag{10}$$

We call this variant GraphDE-a(nalytical) and the optimization procedure in this setting is essentially an EM algorithm [31]: in the E-step, we evaluate the posterior distribution of the latent variable $\mathbf{e}$ by Eq. 10; in the M-step, we optimize the model parameters $\theta$ using gradient descent w.r.t. the objective in Eq. 7. In comparison, GraphDE-v runs more efficiently as it does not need to calculate the posterior, while GraphDE-a can give a tight approximation $\log p_\theta(A, y|X) = \mathcal{L}_{\text{ELBO}}$. We compare the space/time complexity of the algorithms in Appendix F.3.

#### 3.2.2 Instantiations of the Structure Estimation Model

Recall that $p_\theta(\mathbf{A}|\mathbf{X}, \mathbf{e})$ is defined by a two-component mixture (Eq. 5), we need to specify $p_\theta(\mathbf{A}|\mathbf{X})$ and $p_0(\mathbf{A}|\mathbf{X})$ separately for the structure estimation model. We follow the common practice and assume that the edges are conditionally independent [28]. Then the conditional probability of $\mathbf{A}$ can be further factorized as $p_\theta(\mathbf{A}|\mathbf{X}) = \prod_{v,u \in V} p_\theta(\mathbf{a}_{vu}|\mathbf{X})$.

We next consider two versions for $p_\theta(\mathbf{a}_{vu}|\mathbf{X})$. First, we adopt the latent space models (LSM) [15] where the probability of $\mathbf{a}_{vu}$ only depends on the representations of nodes $v$ and $u$:

$$p_\theta(\mathbf{a}_{vu} = 1|\mathbf{x}_v, \mathbf{x}_u) = \sigma([(\boldsymbol{U}\mathbf{x}_v)^\top, (\boldsymbol{U}\mathbf{x}_u)^\top]\boldsymbol{w}). \tag{11}$$

Eq. 11 can be viewed as a logistic regression model where $\sigma(\cdot)$ denotes the Sigmoid function, $\boldsymbol{U}$ is a linear transformation matrix, and $\boldsymbol{w}$ is a learnable vector.

Second, we propose a variant of LSM referred as CosLSM which utilizes the cosine similarity function to measure the similarity between the node representation pair $(\mathbf{x}_v, \mathbf{x}_u)$:

$$p_\theta(\mathbf{a}_{vu} = 1|\mathbf{x}_v, \mathbf{x}_u) = \frac{1}{m} \sum_{i=1}^m \frac{\cos(\boldsymbol{U}_i\mathbf{x}_v, \boldsymbol{U}_i\mathbf{x}_u) + 1}{2}, \tag{12}$$

where we harness multi-head [42] linear transformations $\{\boldsymbol{U}_i\}_{i=1}^m$ for better capacity. Note that if we train the structure estimation model completely based on the observed edges, the model will yield

trivial solutions which predict all edges' probabilities as 1. To avoid this issue, we randomly sample one disconnected node pair labeled as $a_{vu} = 0$ for each existing edge during training.

Finally, for $p_0(\mathbf{a}|\mathbf{X})$, we simply fix it as $p_0(\mathbf{a}_{vu} = 1|\mathbf{X}) = 1/2$. Although more complex distributions can be used for improving the capacity, our experiment results empirically show that this simple distribution is enough for a wide range of debiasing and detection tasks on graphs.

### 3.2.3 Instantiations of the Classification Model

We next specify $p_\theta(\mathbf{y}|\mathbf{A}, \mathbf{X})$ and $p_0(\mathbf{y}|\mathbf{A}, \mathbf{X})$ for the classification model. Similar to the structure estimation model, we fix $p_0(\mathbf{y}|\mathbf{A}, \mathbf{X}) = K^{-1}$ where $K$ denotes the total number of classes. Typically, a GNN takes $(\mathbf{A}, \mathbf{X})$ as input and outputs the probability of $\mathbf{y}$. So for $p_\theta(\mathbf{y}|\mathbf{A}, \mathbf{X})$, we adopt two types of GNN models: the global pooling models [17, 43] and hierarchical pooling models [21, 18]. Denote node $v$'s neighborhoods by $\mathcal{N}_v$, specifically, common GNN convolutional layers execute recursive feature propagation along $G$:

$$\mathbf{x}_v^{(l)} = \sigma\left(\boldsymbol{W}^{(l)} \text{AGG}^{(l)}\left(\{\mathbf{x}_u^{(l-1)}|u \in \mathcal{N}_v \cup \{v\}\}\right)\right), \mathbf{X}^{(0)} = \mathbf{X}, \tag{13}$$

where $\boldsymbol{W}^{(l)}$ denotes weight matrix in the $l$-th layer and $\text{AGG}^{(l)}$ is a permutation-invariant function that aggregates node representations. In hierarchical pooling GNNs, each convolutional layer is followed by a pooling layer to filter out nodes for the next convolution operation:

$$\mathbf{X}^{(l)} = \mathbf{X}_{\text{idx}}^{(l)}, \mathbf{A}^{(l)} = \mathbf{A}_{\text{idx,idx}}^{(l)}, \tag{14}$$

where *idx* is the selected node indices obtained by operations such as selecting the top-$k$ score (calculated by a learnable function) nodes. Finally, for both types of GNNs, a global pooling layer (e.g. max or mean operation over node features) is adopted to extract a graph-level representation $\mathbf{h}$ to be fed into the classifier $\Phi$ whose probability is obtained by an output softmax layer:

$$p_\theta(\mathbf{y}|\mathbf{A}, \mathbf{X}) = \text{Softmax}(\Phi(\mathbf{h})), \mathbf{h} = \text{GlobalPool}(\mathbf{X}^{(n)}), \tag{15}$$

where $n$ denotes the total number of convolutional layers.

## 4 Experiments

In this section, we conduct extensive experiments to answer the following questions:

- **Q1:** How effective is GraphDE in debiased learning and OOD detection?
- **Q2:** How do the composed modules and hyperparameters impact GraphDE's performance? Besides, how does GraphDE take effect during the training procedure?

### 4.1 Experiment Setup

**Datasets.** One synthetic and three real-world datasets of different OOD types are used for debiasing and OOD detection tasks. Here we briefly introduce them, leaving details of statistics to Appendix E.

- **Spurious-Motif** [50] (SPMotif) is a synthetic dataset in which each graph is composed of one base (denoted by $S$) and one motif (denoted by $C$). The graph label $y$ is solely determined by the motif $C$. To simulate distribution shifts, we manually construct spurious correlations between $S$ and $y$. We use datasets with **different spurious correlations** to represent ID and OOD data, respectively. This dataset is unsuitable for typical OOD detection task, as the distribution shift is dominated by label information, opposed to that the OOD detectors focus on detecting input graphs with abnormal structures or node features.
- **MNIST-75sp** [18] converts the MNIST images into super-pixel graphs with at most 75 nodes. For OOD data, we add Gaussian noise to **node features** to simulate distribution shift.
- **Collab** [47] is a social network derived from 3 public collaboration datasets. We use the **graph size** to simulate distribution shift, with graphs of no more than 45 nodes denoted as ID data.
- **DrugOOD** [16] aims to benchmark the OOD generalization performance in the field of AI-aided drug discovery. We use the provided dataset curator to generate a dataset with **different scaffolds**, and treat the largest 6000 molecules as ID data.

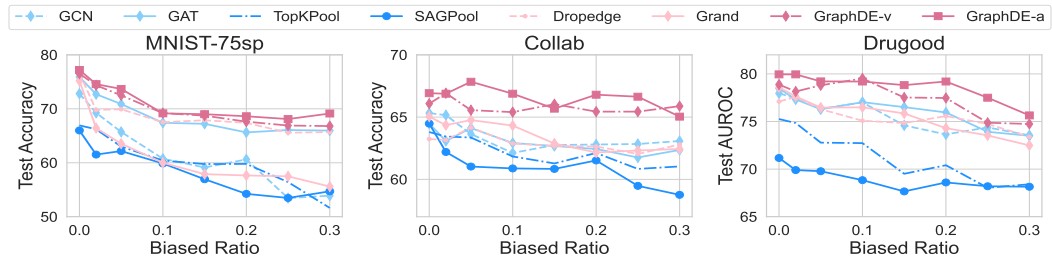

Figure 3: Debiasing results on real-world datasets. The test mean accuracy (or AUROC for DrugOOD) on the ID dataset is plotted. GAT is adopted as the backbone.

Table 1: Debiasing results on SPMotif. It records the test mean accuracy and standard deviation on the ID dataset. The biased ratio is fixed as 10% for fair comparison. Best results are in bold.

| Model | Backbone | DropEdge | GRAND | GraphDE-v | GraphDE-a |
|---|---|---|---|---|---|
| GCN [17] | 68.67±5.04 | 65.59±3.16 | 73.92±3.37 | 75.10±3.36 | **76.59±1.92** |
| GAT [43] | 65.02±3.31 | 62.81±0.64 | 65.27±3.90 | 66.88±1.30 | **67.52±1.84** |
| TopKPool [18] | 78.43±4.93 | 82.32±3.85 | 83.25±3.89 | 85.11±4.42 | **88.54±1.72** |
| SAGPool [21] | 83.57±4.15 | 85.72±3.15 | 86.71±3.49 | 88.96±2.66 | **89.41±2.87** |

Note that during the debiased training stage of GraphDE, we need to mixed a certain number of OOD data into the training dataset. To prevent potential data leakage [14] during the evaluation of OOD detection, the OOD testing dataset should be drawn from a different distribution w.r.t. the OOD data mixed into the training dataset. We denote these two sources of OOD data as the OOD testing dataset and the OOD mixed dataset, respectively. We leave more details of dataset splits to Appendix E.

**GNN backbones.** We adopt two types of GNN backbones for debiasing and OOD detection tasks.

- **Global pooling backbones.** We adopt two popular GNNs: GCN [17] and GAT [43]. To extract the global graph representation for graph classification, we use a global mean pooling or global max pooling layer after the GNNs to summarize the node representations.

- **Hierarchical pooling backbones.** This architecture is comprised of blocks each of which consists of a graph convolutional layer and a pooling layer. Typically, TopKPool [18] selects local parts of input graph for propagation by a learnable score function; SAGPool [21] uses the self-attention mechanism to calculate attention scores, and retain important nodes for propagation.

**Debiasing settings.** For the debiasing task, we mix the training set with certain ratio of outliers and compare models' performance on ID testing data. We evaluate GraphDE's debiasing performance on the four datasets. Besides, As far as we know, there are little existing works concerning debiased learning on graph data. For demonstrating the empirical superiority of our approach, we adopt two recently proposed methods that claim to enhance the general-purpose robustness of GNNs as competitors: [3]: DropEdge [39] randomly removes a number of edges from the input graph in each training epoch, and GRAND [11] uses a random propagation strategy to perform graph augmentation. We study the negative effect of outliers by varying the **biased ratio** ($|\mathcal{D}_{out}^{tr}|/|\mathcal{D}^{tr}|$) of training data.

**OOD detection settings.** We evaluate GraphDE's OOD detection performance on three real-world datasets (with equal number of ID and OOD testing samples), using three widely used metrics [14]: area under the receiver operating characteristic curve (*AUROC*), area under the precision-recall curve (*AUPR*), and the false positive rate at 95% true positive rate (*FPR95*). Besides, we consider three types of baselines: first, we use the backbones' max softmax score (MSP) [13] as a vanilla baseline; second, we adopt two-stage graph kernel baselines in [52] (specifically, we adopt WL [40] and PK [34] kernels, LOF [5] and OCSVM [6] detectors); finally, we use OCGIN [52] and GLocalKD [29] as two deep learning based detection baselines.

---

[3]Notably, they can also be integrated with GraphDE to achieve even better debiasing performance.

Table 2: Detection results on real-world datasets. The biased ratio is fixed as 30% for fair comparison. We report the mean and standard deviation for all the detectors, except the two-stage models, which are invariant to different random seeds. We use the GraphDE-a variant as the OOD detector.

| OOD Detector | MNIST-75sp | | | Collab | | | DrugOOD | | |
|---|---|---|---|---|---|---|---|---|---|
| | AUROC↑ | AUPR↑ | FPR95↓ | AUROC↑ | AUPR↑ | FPR95↓ | AUROC↑ | AUPR↑ | FPR95↓ |
| MSP [13] | 62.37±2.96 | 60.71±1.83 | 88.60±2.71 | 51.37±4.24 | 53.19±3.68 | 91.00±2.41 | 54.57±9.18 | 52.43±6.85 | 90.76±4.95 |
| WL [40]+OCSVM [6] | 75.35 | 60.72 | 32.75 | 64.61 | 60.39 | 64.80 | 66.84 | 72.41 | 81.72 |
| WL [40]+LOF [5] | 61.62 | 57.22 | 94.20 | 67.72 | 62.64 | 81.40 | 56.21 | 49.33 | 80.80 |
| PK [34]+OCSVM [6] | 72.26 | 59.95 | 47.80 | 64.57 | 62.19 | 70.60 | 66.08 | 61.02 | 82.60 |
| PK [34]+LOF [5] | 61.19 | 58.51 | 92.55 | 64.25 | 58.76 | 91.20 | 57.40 | 51.42 | 88.20 |
| OCGIN [52] | 65.07±2.55 | 60.13±2.45 | 77.39±5.55 | 70.48±2.72 | 71.77±1.84 | 86.70±0.71 | 68.39±4.77 | 66.05±5.11 | 82.80±7.50 |
| GLocalKD [29] | 86.22±0.78 | 87.59±0.87 | 25.72±5.90 | 71.82±0.47 | **72.91±0.24** | 70.04±1.32 | 63.42±0.60 | 58.03±0.64 | **70.28±1.83** |
| GraphDE | **95.53±3.63** | **94.78±5.09** | **19.24±9.33** | **72.15±2.27** | 71.86±2.54 | **64.40±0.41** | **69.15±1.11** | **67.40±0.51** | 80.30±0.33 |

## 4.2 Main Results (Q1)

We train and evaluate each model under the settings as described in the last section (more hyperparameter and training/testing details in Appendix D). All the experiments are run over 5 different random seeds. More experimental results can be found in Appendix F. We make the following observations:

**Debiasing results.** Tab. 1 reports GraphDE's debiasing performance on SPMotif. In this setting, we treat the robust GNN baselines and two variants of our GraphDE as plug-in modules and apply them to all the four backbones for comparison. The results show that GraphDE consistently (for all the backbones) outperforms the baselines by a large margin along with low variance. Specifically, for the TopKPool backbone, GraphDE-a surpasses the backbone by 10.11% and the strongest baseline (GRAND) by 5.29%. Such improvements strongly suggest that GraphDE can achieve better debiasing performance for various backbones. Besides, the results also show that the GraphDE-a generally outperforms GraphDE-v, as the EM algorithm can give a tight approximation for the log-likelihood.

Fig. 3 illustrates GraphDE's performance on three real-world datasets. In this setting, we mainly study the debiasing performance w.r.t. different biased ratios, so the backbone for robust GNN baselines and GraphDE are all fixed as GAT. As shown in the figure, GraphDE consistently outperforms baselines on the three datasets for all the biased ratios. The backbones' performance drops drastically as the biased ratio gets higher, and the robust GNN baselines can mitigate this effect to some extend. In comparison, GraphDE performs much more stable. This advantage is more pronounced when the biased ratio is large, e.g. GraphDE surpasses all the other baselines by nearly 5% on Collab when the biased ratio is 0.2. An interesting observation is that GraphDE can also achieve a better test accuracy when the biased ratio is 0, implying that the pure training set also violates the i.i.d. assumption.

**Detection results.** Tab. 2 reports GraphDE's detection performance on three real-world datasets. As can be seen from the table, the max softmax score (MSP) achieves low detection performance. It suggests that vanilla GNNs are poorly calibrated [35] and it is of critical need to develop new graph-level OOD detectors. The results show that GraphDE performs the best on 7 out of 9 benchmarks. Specifically, on MNIST-75sp, GraphDE surpasses the strongest baseline by approximately 10% for AUROC, while also pulls down the false positive rate (FPR95).

## 4.3 Further Study (Q2)

**Impact of prior ratio.** We select different prior ratios $1 - p(\mathbf{e})$ to study its impact on debiasing performance. As shown in the Fig. 4(a), the ID test accuracy doesn't fluctuate much from 0.05 to 0.7, but drops drastically when prior ratio goes higher than 0.7. Besides, we also find that the test accuracy peaks at prior ratio 0.1 for biased ratio 0.1, and 0.2 for biased ratio 0.2, exactly when $\mathcal{L}_{kl}$ becomes 0.

**Ablation on $\mathcal{L}_{reg}$.** We remove $\mathcal{L}_{reg}$ in the objective to study the structure estimation model's impact on debiasing performance. As illustrated in Fig. 4(b), we can find that the test accuracy drops obviously at all biased ratios. Specifically, The test accuracy drops over 6% at biased ratio 0.25. The results prove the structure estimation model's capability of down-weighting outliers in training data.

**Ablation on $\mathcal{L}_{cls}$.** We remove $\mathcal{L}_{cls}$ to study the influence of the classification model on the OOD detection performance. As shown in Fig. 4(c), we plot the detection AUROC on the testing dataset w.r.t. different biased ratios. The figure illustrates that the AUROC generally drops as the biased ratio gets higher, since the outliers will hurt the fitting performance of the structure estimation model. Typically, we can see that the blue line (without $\mathcal{L}_{cls}$) is consistently lower than the red line, with the

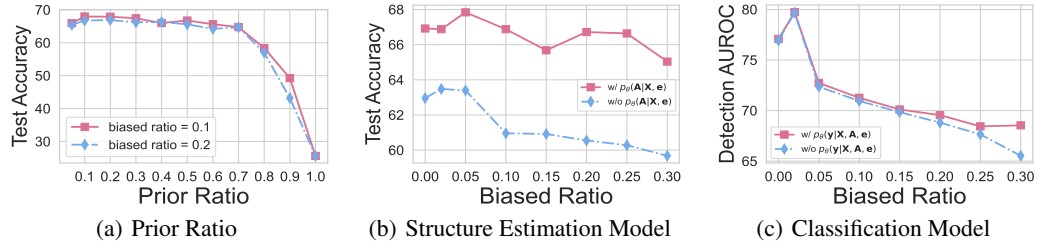

Figure 4: Sensitivity analysis and ablation studies (on Collab). We use the GraphDE-a variant.

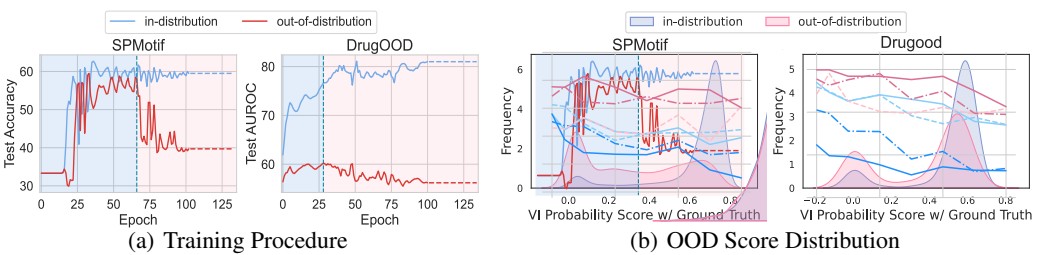

Figure 5: Explanatory experiments. We use the GraphDE-a variant and the biased ratio is set as 0.2.

gap getting larger as the biased ratio goes higher. Specifically, the performance gap is approximately 3% when biased ratio is 0.3. The results prove that the classification model can help down-weight training outliers, thereby promoting the detection performance of the structure estimation model.

**Training procedure.** We plot GraphDE's test accuracy after every training epoch on ID and OOD (i.i.d. to training outliers) datasets. As shown in Fig. 5(a), the ID test accuracy simply fluctuates, rises, and converges. In comparison, the trend of OOD test accuracy can be divided into two phases. As separated by the green dashed line, we can see the red line first increases then decreases. Generally, in the increasing phase, GraphDE is optimized to better fit the outliers in the training data. However, in the decreasing phase, as the outliers are down-weighted by a large extend, GraphDE focuses on fitting ID data. This transition prevents overfitting of outliers and promotes the ID test accuracy.

**Visualization of inferred probability.** We visualize the recognition model's inferred probability $q_\phi(e = 1|A, X, y)$ on the training dataset. As shown in Fig. 5(b), ID data is assigned with a higher probability on both datasets. In particular, most outliers are assigned with probability at around 0 on SPMotif. This directly indicates that the outliers are down-weighted during the training procedure.

## 5 Conclusion and Outlook

In this paper, we have provided a viable approach to the problem of debiased learning and OOD detection over graph data. Specifically, a generative process is modeled to capture distribution shifts of graph data. By introducing a variational recognition model to infer the environment variable and two-component mixed generative models, the learning objective for GraphDE is derived, which can identify and down-weight outliers during training, as well as inducing an effective OOD detector on new data. While our framework is mainly designed for graph data where both the debiased learning and OOD detection remain relatively under-explored, it can be potentially generalized to other data formats like images or texts, which we leave for future studies.

## Acknowledgement

This work was partly supported by National Key Research and Development Program of China (2020AAA0107600), National Natural Science Foundation of China (61972250, 72061127003), and Shanghai Municipal Science and Technology (Major) Project (2021SHZDZX0102, 22511105100).

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
