# OpenReview forum: "GraphDE: A Generative Framework for Debiased Learning and Out-of-Distribution Detection on Graphs"
_NeurIPS.cc/2022/Conference — NeurIPS 2022 Accept_

### Official Review · Reviewer_hmWz · 2022-07-07

**Rating:** 6
**Confidence:** 3
**Soundness:** 2 fair
**Presentation:** 4 excellent
**Contribution:** 3 good

**Summary:**

This paper proposes a framework called GraphDE, which unifies debiased learning and OOD detection for graph data. GraphDE takes a generative view to model the joint distribution of $\{G, y, e\}$, with an additional recognition model that infers an OODness indicator variable $e$. The loss is carefully designed according to the framework, with interpretable justification. Various and extensive experiments demonstrate the effectiveness of GraphDE, which is expected to provide reference for future related works.

**Questions:**

I have a few confusions. Please correct me if I am wrong:
1. How do you calculate the threshold in Eq. (3)?
2. Line 107-108, the authors mention that prior works are limited to feature-label shift but ignore the variation in the feature space. Can GraphDE capture the variation in the feature space?
3. Line 132, why are the two distributions parameterized by the same $\theta$?
4. Line 134, I think there is a typo, i.e. $p_\theta(X|A,e)$ should be $p_\theta(A|X,e)$.
5. In the last term of Eq. (7), should $p(e)$ be $p(e_i)$?
6. I am very confused with line 181-182 which mentions that $p(e)$ is a scalar $\in[0,1]$. If so, why $p(e)$ is not canceled out in Eq. (8)? I think it should be a discrete distribution over ID and OOD input.
7.  Do you have the assumption that training and test data have the same $p(e)$? Because $p(e)$ (I suppose it is obtained in the training set) is used in the inference on test data (in Eq.(8)). If test data have a different $p(e)$, will GraphDE still work?
8. Eq.(9) is used to predict label on test data. Do you still output a label even when OOD is detected?
9. What is the role of GraphDE-v? Since you can compute the posterior analytically, it seems to me GraphDE-a works perfectly, and it is also confirmed in Table 1 that GraphDE-v is entirely outperformed by GraphDE-a.
10. Figure 4 is counter-intuitive to me. Eq.(2) tells me that debiased learning can be reduced to ignoring updates from outliers (because training on outliers will hurt the performance), and proof in Appendix A.2 tells me that optimizing the proposed $L_{cl}$ is equivalent to optimizing Eq.(2). Then I suppose $L_{cl}$ is why GraphDE can boost test accuracy as in Figure 1(a). However, the ablation study shows that removing $L_{cl}$ makes literally no difference to the performance, while removing $L_{reg}$ shows more impact. Do you have any interpretation on this result?

**Ethics Review Area:**

["I don’t know"]

**Limitations:**

The authors addressed the limitations and societal impact in their paper (in the end of appendix).

**Strengths And Weaknesses:**

Strength:
1. The framework unifies debiased learning and OOD detection, i.e. the OOD detector is obtained during the debiased learning.
2. The loss is well-interpretable.
3. Experimental results are fairly good.


Weakness:
See my questions below.

---

> ### Author Response · Authors · 2022-08-02
> **Response to reviewer hmWz.**
>
> Thank you for the positive feedbacks about our framework, theory, and experiments. Hope the following responses can help relieve your concerns:
>
> **Q1:** How do we calculate the threshold in Eq. (3)?
>
> **R1:** Eq. (3) in our paper defines the pratically used OOD detector with a threshold $\tau$. However, we emphasize that there is no need to specify $\tau$ beforehand to evaluate the detector's performance. As stated in Section 4.1 and shown in Table 2, we adopt three widely used metrics: **AUROC**, **AUPR**, and **FPR95** to measure the OOD detection performance. For AUROC and AUPR, we continuously change $\tau$ to get the ROC curve and PR curve of the testing dataset, then calculate the area under them as the measure. For FPR95, we adjust $\tau$ to get 95% true positive rate and compute the corresponding false positive rate. So in this paper, we have no need to calculate the threshold. To get an OOD detector for practical use, for example, we can choose $\tau$ that misclassifies the minimum number of testing ID and OOD data as the threshold.
>
> **Q2:** Can GraphDE capture the variation in the feature space?
>
> **R2:** Yes, different from data resampling methods [1] [2] that completely rely on the supervised loss to identify and down-weight outliers in the training set (the same role as our classification module $p_\theta(\mathbf y|\mathbf X,\mathbf A,\mathbf e)$), we additionally introduce an structure estimation module $p_\theta(\mathbf A|\mathbf X,\mathbf e)$, which plays an important role in the learning objective as shown in Eq. (7). In this sense, as proved by our theory, GraphDE can also automatically learn to detect and down-weight those training samples with abnormal relationship between the adjacency matrix $\mathbf A$ and node features $\mathbf X$. This directly enables GraphDE to capture variation in the feature space.
>
> **Q3:** Line 132, why are the two distributions parameterized by the same $\theta$?
>
> **R3:** We use one parameter $\theta$ to model the classification module $p_\theta(\mathbf y|\mathbf X,\mathbf A,\mathbf e)$ and the structure estimation module $p_\theta(\mathbf A|\mathbf X,\mathbf e)$ for notation simplicity. You can interpret it as $\theta=[\theta_{cl},\theta_{reg}]$, where $\theta_{cl}$ controls the distribution $p_\theta(\mathbf y|\mathbf X,\mathbf A,\mathbf e)$, while $\theta_{reg}$ determines $p_\theta(\mathbf A|\mathbf X,\mathbf e)$.
>
> **Q4:** There is a typo in line 134 that $p_\theta(\mathbf X|\mathbf A,\mathbf e)$ should be $p_\theta(\mathbf A|\mathbf X,\mathbf e)$.
>
> **R4:** Thank you, this is indeed a typo and it has been fixed in the updated version.
>
> **Q5:** In the last term of Eq. (7), should $p(\mathbf e)$ be $p(\mathbf e_i)$?
>
> **R5:** Specifically, Eq. (7) is the sample-wise unfolded form of Eq. (6). And you can see from it that the ELBO includes the KL-divergence between the posterior distribution and prior distribution of the environmental variable $\mathbf e$. Therefore, $p(\mathbf e)$ represents our prior knowledge about the dataset, which is **shared over all the data points** [3]. So there is no need to use one $p(\mathbf e_i)$ for each training sample.
>
> **Q6:** Why line 181-182 mentions that $p(\mathbf e)$ is a scalar $\in$ [0, 1] but not a discrete distribution over ID and OOD input?
>
> **R6:** We're sorry for the incorrect expression. In fact, as you said, $p(\mathbf e)$ is a Bernoulli distribution over the environment variable. In this sense, we have $p(\mathbf e)=\text{Bernoulli}(\alpha)$ where $\alpha\in[0, 1]$ represents the portion of ID data in the dataset. We have fixed this issue in the updated version.
>
> **Q7:** Do we assume that training and testing data have the same $p(\mathbf e)$? If test data have a different $p(\mathbf e)$, will GraphDE still work?
>
> **R7:** We point out that $p(\mathbf e)$ stands for the prior distribution under the probabilistic framework, and it is shared across training and testing data since it encodes our belief about the probability a data point is ID or OOD. If testing data have a different $p(\mathbf e)$, GraphDE will still work since we actually compute a posterior distribution over the environmental variable using Eq. (8). Thus, it can distinguish those OOD samples with abnormal adjacency matrix $\mathbf A$ or node features $\mathbf X$.
>
> **Q8:** Do we still output a label when OOD is detected?
>
> **R8:** No, we just report an OOD warning and reject to predict on these samples, which is the same as done in a number of related works [4] [5].

---

> > ### Author Response · Authors · 2022-08-02
> > **Response to reviewer hmWz (cont.)**
> >
> > **Q9:** What is the role of GraphDE-v? It seems that GraphDE-v is entirely outperformed by GraphDE-a.
> >
> > **R9:** Thank you for pointing out our negligence. As we have discussed in Section 3.2.1 (Instantiations of the Recognition Model), GraphDE-v directly assigns a learnable scalar for each sample in the training dataset, i.e. $q_\phi(\mathbf e_i|A_i, X_i, y_i)=\text{Bernoulli}(\alpha_i)$ where $\alpha_i\in[0,1]$ is a learnable parameter. For GraphDE-a, however, we need to compute the posterior analytically using Eq. (10). Therefore, GraphDE-a owns a substantial larger time and spatial complexity than GraphDE-v, and the latter is also easier to be implemented. Though GraphDE-a gives a tight approximation and performs better, GraphDE-v may be a considerable choice when we have limited space/time resource.
> >
> > **Q10:** Figure 4 is somewhat counter-intuitive. Why the ablation study shows that removing $\mathcal L_{cl}$ makes literally no difference on the performance, while removing $\mathcal L_{reg}$ shows more impact?
> >
> > **R10:** It seems that you are confused with our Figure 4.(b) and Figure 4.(c). In Figure 4.(b), the test accuracy drops significantly after we remove $\mathcal L_{reg}$, this result is consistent with our theory that the structure estimation module acts as a regularization to detect and down-weight the outliers in the training dataset. In Figure 4.(c), as discussed in Section 4.3, we **measure the detection AUROC but not the test accuracy** after removing  $\mathcal L_{cls}$. Note that we cannot even conduct label prediction without the classification module (i.e. $\mathcal L_{cls}$). Therefore, these two figures **cannot be compared with each other since they indeed compute a different measure**. The detection performance degradation in Figure 4.(c) is consistent with our theory that the classification and the structure estimation module are learned in a mutually-promote manner.
> >
> >
> > ### References
> >
> > [1] Training Deep Neural Networks on Noisy Labels with Bootstrapping, ICLR 2015.
> >
> > [2] Learning to Reweight Examples for Robust Deep Learning, ICML 2018.
> >
> > [3] Advances in Variational Inference, TPAMI 2019.
> >
> > [4] Deep Anomaly Detection with Outlier Exposure, ICLR 2019.
> >
> > [5] Energy-based Out-of-distribution Detection, NeurIPS 2020.

---

> > > ### Comment · Reviewer_hmWz · 2022-08-07
> > > **Updated review**
> > >
> > > Thank the authors for their response.  The response confirms that the initial draft lacks some clarity, but most of my concerns are addressed in the rebuttal.
> > >
> > > Honestly,  my expertise in GNN OOD detection is limited. The novelty claimed by the authors looks good to me, but I am not so sure whether they overclaim or not compared to SOTA related works, and also not sure whether the experiment is thorough or fair enough. I tend to not change my initial rating, but I am open to opinions from other reviewers or ACs.

---

> > > > ### Author Response · Authors · 2022-08-09
> > > > **Thank you**
> > > >
> > > > Thank you for the positive feedbacks. According to your suggestions, we have addressed the clarity problems in the revised version. These updates undoubtedly improve the paper.

---

### Official Review · Reviewer_xAwj · 2022-07-11

**Rating:** 4
**Confidence:** 4
**Soundness:** 3 good
**Presentation:** 2 fair
**Contribution:** 2 fair

**Summary:**

This paper aims to address two challenging tasks, i.e. 1) learning debased GNN model from the training data with OOD samples and 2) detecting the OOD samples from testing data, with a unified framework. To this end, a novel method named GraphDE is produced on the basis of variational inference. A probabilistic generative model is introduced to model the distribution of ID and OOD samples, and the classification module and OOD detection module are integrated into the learning framework. Experiments are conducted to verify the performance of GraphDE on both debiased learning and OOD detection tasks.


**Questions:**

- Is that practical to define the larger graphs in a dataset (i.e. Collab) or the images with gaussian noise (i.e. MNIST-75sp) as the OOD samples? It is hard to say these samples are generated from a different distribution.
- In Line 134, the authors say "p_\theta(X|A,e)" is composed of an ID component and an OOD component. Is there a typo? Because the assumption is that A is generated from X.


**Limitations:**

The authors have provided the limitations and potential negative impacts of the proposed framework in Appendix G and H, respectively. However, more discussion for potential negative impacts should be given, for example, the debiased learning paradigm may cause the concern about fairness.

**Strengths And Weaknesses:**

Pros:
- The research problem is interesting.
Both debiased graph learning and graph OOD detection are attractive and interesting topics in the research community. The authors discuss the similarities and differences of these two tasks and address both tasks with a unified learning framework, which is promising and challenging.
- The results are promising.
In the experiment, the authors conduct quite extensive experiments, and the proposed method achieves very good results, which prove its effectiveness.
Cons:
- The construction of graph generative model is defective.
In the generative model, the distribution of feature p(X) is not modelled specifically, and the generation of feature X is not involved in the generating process. However, the ID and OOD data obviously have different feature distributions. For example, molecules from different domains tend to be formed by different atoms. In this case, it is not practical to model ID/OOD graph data without considering the difference in feature distribution.
- The instantiation of OOD structure distribution p_0 is not practical.
In the instantiation, the authors use a simple distribution p(a=1)=1/2 to model the OOD data. However, in practice, the real-world OOD data can be universal, which cannot be simply modelled by such an impracticable distribution. Moreover, the ID data is also possible to follow "p(a=1)=1/2". In this case, how to distinguish ID/OOD distributions in practice? Therefore, I believe modelling OOD data with more complex or with learnable distribution is more persuasive in this method.
- There is potential leakage in the evaluation of OOD detection.
Since the model is trained on data composed by both ID and OOD data, the patterns of OOD data have already been seen by the model. In this way, a concern raises that the knowledge about OOD data would leak during the learning procedure. Such leakage may lead to unfair comparison in evaluation.
- More baseline methods should be considered for comprehensive comparison.
Also graph debiased learning and graph OOD detection are both new-born directions in graph learning, there are already some pioneering works that focus on these topics. However, the authors consider limited baselines for comparison, which reduces the persuasion of experiments. So, more baselines can be consider for debiased learning (e.g., [*1], [*2]) and OOD detection (e.g., [*3]). More related papers can be found in survey [*4].
[*1] Fan, Shaohua, et al. "Generalizing Graph Neural Networks on Out-Of-Distribution Graphs." arXiv preprint arXiv:2111.10657 (2021).
[*2] Li, Haoyang, et al. "Ood-gnn: Out-of-distribution generalized graph neural network." arXiv preprint arXiv:2112.03806 (2021).
[*3] Ma, Rongrong, et al. "Deep Graph-level Anomaly Detection by Glocal Knowledge Distillation." Proceedings of the Fifteenth ACM International Conference on Web Search and Data Mining. 2022.
[*4] Li, Haoyang, et al. "Out-of-distribution generalization on graphs: A survey." arXiv preprint arXiv:2202.07987 (2022).

---

> ### Author Response · Authors · 2022-08-02
> **Response to reviewer xAwj.**
>
> Thank you for your time and valuable suggestions. We are glad that you appreciate our topic, methodology, and experiments. We also add new experiment results and explanations in the hope that they can address your concerns:
>
> **Q1:** The distribution of features $p(\mathbf X)$ should be modelled since ID and OOD data can obviously have different feature distributions.
>
> **R1:** Thank you for proposing this important question. It is undoubted that ID and OOD data can have different feature distributions and we need to capture this difference to seperate them out for better debiasing/detection performance. However, $p(\mathbf X)$ is not necessarily to be modelled since the conditional distribution $p(\mathbf A|\mathbf X)$ is a function of ajacency matrix and node features, i.e. it can capture both the distribution shifts of $\mathbf A$ and $\mathbf X$. Besides, as shown in Eq. (6), we are maximizing the log probability conditional on $X$ so there is no need to model $p(\mathbf X)$. This technique has been widely adopted in the graph learning community [1] [2], perhaps both for its simplicity, and for that there rarely exists suitable model for the node feature distribution $p(\mathbf X)$. We will leave further study for future work.
>
> **Q2:** The instantiation of OOD structure $p_0$ with a simple $p(a=1)=\frac{1}{2}$ is not practical. More complex or learnable method should be used for it.
>
> **R2:** We agree that we can use more complex or learnable model to instantiate the outlier component. However, we use $p(\mathbf a=1)=\frac{1}{2}$ in this paper for its simple implementation and promising empirical power as proved in the experiment section. For your concern that real-world OOD data are universal and cannot be captured by this simple distribution, we emphasize that **the outlier component does not necessarily need to perfectly fit the OOD distribution.** As we have stated in Proposition 2, GraphDE can learn to assign higher probability for ID data if $p_\theta(\mathbf A|\mathbf X)$ and $p_\theta(\mathbf y|\mathbf A,\mathbf X)$ (the ID components) can better fit ID data than the outliers. This does not add assumptions on the OOD component (but undoubtedly, a better fitted OOD component can help better learn the distribution of environment variable). Also, if the ID data also follows "$p(\mathbf a=1)=\frac{1}{2}$", we can distinguish between ID and OOD data as long as $p_\theta(\mathbf A|\mathbf X)$ can better fit the ID distribution. If ID and OOD data have the same adjacency matrix distribution, then the distribution shifts should only lie in node features $\mathbf X$ and labels $\mathbf y$, which can also be captured by GraphDE.
>
> **Q3:** There is potential leakage in the evaluation of OOD detection.
>
> **R3:** Thank you for raising this considerable question. As what you have said, we make the training outliers and OOD testing samples orthogonal (i.e. do not intersect with each other) this time. The results are shown in the following table (these results will be updated in the final version). Specifically, "MNIST-75sp 0.3,0.6" denotes that we add Gaussian noise with a mean of 0.3 to the training outliers, and Gaussian noise with a mean of 0.6 to the OOD testing samples; "Collab 45, 80, 100" represents that we treat graphs with 45-80 nodes as ID data, 80-100 as training outliers, and those with more than 100 nodes as OOD testing samples. The other settings are kept the same as in the main text. The table shows that GraphDE still outperforms the baselines on 5 out of 6 metrics on these two datasets. Besides, the performance of GraphDE is comparable to or even better than in the original paper, proving the excellent detection power of GraphDE. We will add these part of results to the paper in the revised version.

---

> > ### Author Response · Authors · 2022-08-02
> > **Response to reviewer xAwj (cont.)**
> >
> > | Dataset  |                  | MNIST-75sp 0.3,0.6 |                    |                  | Collab 45,80,100 |                    |
> > | -------- | :--------------: | :----------------: | :----------------: | :--------------: | :--------------: | :----------------: |
> > | Detector | AUROC $\uparrow$ |  AUPR $\uparrow$   | FPR95 $\downarrow$ | AUROC $\uparrow$ | AUPR $\uparrow$  | FPR95 $\downarrow$ |
> > | MSP      |    62.37±2.96    |     60.71±1.83     |     88.60±2.71     |    51.37±4.24    |    53.19±3.68    |     91.00±2.41     |
> > | WL+OCSVM |      75.35       |       60.72        |       32.75        |      64.61       |      60.39       |       64.80        |
> > | WL+LOF   |      61.62       |       57.22        |       94.20        |      67.72       |      62.64       |       81.40        |
> > | PK+OCSVM |      72.26       |       59.95        |       47.80        |      64.57       |      62.19       |       70.60        |
> > | PK+LOF   |      61.19       |       58.51        |       92.55        |      64.25       |      58.76       |       91.20        |
> > | OCGIN    |    65.07±2.55    |     60.13±2.45     |     77.39±5.55     |    70.48±2.72    |  **71.77±1.84**  |     86.70±0.71     |
> > | GraphDE  |  **94.53±4.63**  |   **93.78±5.09**   |   **19.24±9.33**   |  **72.15±2.27**  |    68.46±2.54    |   **64.40±0.41**   |
> >
> > **Q4:** More baseline methods should be considered for comprehensive comparison.
> >
> > **R4:** Thank you for pointing out the baselines for us to compare with. First, we wish to resolve the misunderstanding of relation between [4] [6] and our paper. Specifically, these two papers focus on the topic of OOD generalization, which is orthogonal to our work, as we have discussed in Appendix B. To summarize, OOD generalization aims at training a model that can generalize to the unknown testing distribution from the limited training data. However, debiased learning wishes to identify the outliers (harmful examples) in the training dataset and mitigate their bad effects during training. That's why we do not adopt these baselines in our paper. Next, GLocal [5] is yet another interesting graph OOD detection baseline that we ignored since it is just published in this year's WSDM. Typically, it trains one GNN to predict another GNN with randomly initialized network weights to learn graph representations that capture both local and glocal information of graphs. We adapt its published code (using default hyperparameters) to run on two of our datasets, with the results in the following table. As we can see, GraphDE outperforms GLocal across the 6 metrics on the 2 datasets, which further proves the detection capability of our GraphDE.
> >
> > | Dataset  |                  |   MNIST-75sp    |                    |                  |     Collab      |                    |
> > | -------- | :--------------: | :-------------: | :----------------: | :--------------: | :-------------: | :----------------: |
> > | Detector | AUROC $\uparrow$ | AUPR $\uparrow$ | FPR95 $\downarrow$ | AUROC $\uparrow$ | AUPR $\uparrow$ | FPR95 $\downarrow$ |
> > | GLocal   |    80.53±0.38    |   78.72±1.11    |     62.11±6.43     |    66.64±1.56    |   62.00±2.08    |     73.68±1.77     |
> > | GraphDE  |  **93.14±5.42**  | **92.95±5.33**  |   **29.86±9.54**   |  **70.54±0.34**  | **66.73±0.13**  |   **66.28±1.08**   |
> >
> > **Q5:** Is that practical to define larger graphs in a dataset (i.e. Collab) or the images with gaussian noise (i.e. MNIST-75sp) as the OOD samples? It is hard to say these samples are generated from a different distribution.
> >
> > **R5:** Before our work, there has already been a series of works [3] [4] focus on the OOD generalization problem for GNNs. These works have used the graph size of Collab, and gaussian noise of MNIST-75sp to construct distribution shifts, which is consistent with this paper. Therefore, it's reasonable for us to get OOD samples by these factors. More intuitively, larger graphs have a different distribution of adjacency matrix compared to smaller graphs. Besides, images with gaussian noise will end up with different node features from the original graphs. Notably, [5] proposes to treat the graphs in minor class as the OOD samples. This is also a potential way for us to construct distribution shifts.

---

> > > ### Author Response · Authors · 2022-08-02
> > > **Response to reviewer xAwj (cont.)**
> > >
> > > **Q6:** Is there a typo in line 134? Because the assumption is that $\mathbf A$ is generated from $\mathbf X$.
> > >
> > > **R6:** Thank you, this is indeed a typo and it has been fixed in the updated version.
> > >
> > > Also thank you for pointing out our ignorance on the negative impact of  GraphDE. As we focus on developing trustworthy GNNs, we believe that the negative impacts of our work are small compared to its contributions. However, it can still raise problems like data fairness due to its re-sampling strategy to conduct debiasing. Besides, its robustness as an OOD detector should be studied in-depth as future work, since malicious attackers may fool GraphDE to treat OOD data as ID data, leading to potential performance degradation in practice. We have added these discussions in our newly submitted version.
> > >
> > > ### References
> > >
> > > [1] A Flexible Generative Framework for Graph-based Semi-supervised Learning, NeurIPS 2019.
> > >
> > > [2] Graph Stochastic Neural Networks for Semi-Supervised Learning, NeurIPS 2020.
> > >
> > > [3] Discovering Invariant Rationales for Graph Neural Networks, ICLR 2022.
> > >
> > > [4] OOD-GNN: Out-of-Distribution Generalized Graph Neural Network, Arxiv.
> > >
> > > [5] Deep Graph-level Anomaly Detection by Glocal Knowledge Distillation, WSDM 2022.
> > >
> > > [6] Generalizing Graph Neural Networks on Out-of-Distribution Graphs, Arxiv.

---

> ### Author Response · Authors · 2022-08-09
> **To reviewer xAwj: Looking forward to your reply**
>
> While the other reviewers have acknowledged our rebuttal and raised their rating accordingly, we are wondering whether our responses have addressed your concerns properly. Your feedback will definitely help reach a more reasonable decision on our submission. Thank you!

---

### Official Review · Reviewer_iPSD · 2022-07-11

**Rating:** 3
**Confidence:** 4
**Soundness:** 2 fair
**Presentation:** 1 poor
**Contribution:** 2 fair

**Summary:**

This paper studies debiased learning and OOD detection for GNNs. Specifically, the authors propose GraphDE, a probabilistic generative framework to model the distribution shifts of graph data. The proposed method contains three main modules: the recognition model to infer the environment variables, the structure estimation model to detect outlier and OOD testing data, and the classification GNN model. Theoretical and empirical justifications of the proposed method are provided.

**Questions:**

See above

**Limitations:**

N.A.

**Strengths And Weaknesses:**

Pros
[+] The motivations are clearly present. Figure 1 also helps understand the research problem of this paper.
[+] The proposed method shows improvements in the adopted datasets.
[+] The sensitivity analysis and ablation studies are provided to gain deeper insights into the proposed method.

Cons:
[-] The novelty of the paper is somewhat limited. The essential idea of the proposed method is building a variational inference module on top of the existing GNNs, which are both heavily studied in the literature. Besides, the variational inference to infer the environment variable seems rather general and hardly connected with graph data (i.e., it may also be applied to other data types). However, the authors claim the challenges of non-Euclidean graph data in the introduction.

[-] The authors do not provide time complexity analysis or the time cost in practice, so the efficiency aspect of the proposed method is unclear.

[-] The experiments are not entirely convincing.
a) Most importantly, the authors do not compare with OOD baselines, including general OOD methods (e.g., IRM, DRO, etc.), which can be directly combined with GNN backbones, and recent methods designed explicitly for graphs (e.g., DIR and OOD-GNN).
b) The authors only consider GCN and GAT as the backbone but ignore more popular and powerful backbones such as GIN.
c) Though I acknowledge that some real-world graph benchmarks are adopted, it would make the experiments more convincing if OGB datasets, widely adopted in the literature, are further included.

[-] The writing of the paper could be improved:
a) Prop 4.1 and 4.2 are in plain language and thus may not be rigorous. It would be better if these propositions are provided in mathematical terms.
b) Some of the claims are not well supported. For example, one of the claimed contributions is that the modules are optimized in a mutually-promoting manner, which is not verified in the experiments. The necessary discussions on the inferred environments during the training of GNNs (as stated in line 12) have also not been present in detail.
c) The problem formulation in Section 2 seems to be literally identical to the existing work EERM, e.g., lines 69-73 and Section 2.1 in EERM. Such textual overlap without quotation should be avoided.

[-] The authors claim in line 109: “For the first time to our best knowledge, we formally define and deal with the graph debiased learning problem.”, which misses important related works such as [1-2].
[1] Shift-Robust GNNs: Overcoming the Limitations of Localized Graph Training Data, NeurIPS 2021
[2] Debiased Graph Neural Networks with Agnostic Label Selection Bias, TNNLS 2022

Minor:
(1) More related works in Appendix B seem to highly relate to the paper and should be moved to the main paper.
(2) The font of “idx” in line 222 should be kept consistent with that in Eq. (14).

---

> ### Author Response · Authors · 2022-08-02
> **Response to reviewer iPSD.**
>
> Thank you for your time and thorough reviews. We are happy that you appreciate our motivation, methodology, and in-depth studies. Here are our responses to your problems:
>
> **Q1:** The novelty of the paper is somewhat limited as VI and GNNs are both heavily studied in the literature. The method therefore seems hardly connected with graph data.
>
> **R1:** We agree that VI and GNNs are two fields that have been deeply studied in the literature. However, we emphasize that **this paper's main novelty does not lie in these two conventional methodologies**. As we have claimed in Section 1, in this paper, we focus on the debiasing and OOD detection problems for GNNs: 1) Most importantly, we model the graph generative process and therefore propose a unified probabilistic framework to define and tackle these two problems simultaneously. 2) Besides, through introducing a two-component structure for the generative models, we induce a novel learning objective and justify that GraphDE can identify and down-weight outliers during training while provide an OOD detector on test set. For your concern that the method seems hardly connected with graph data, similar to works [1] [2] that utilizes generative perspectives to model graph data, **our generative process modelling is tightly correlated with graphs**. Potentially this method can be extended to dealing with CV or NLP datasets (we regard it as a strength rather than a weakness), but this may need non-trivial efforts to re-formulate the generative and recognition models.
>
> **Q2:** The authors should provide time complexity analysis or the time cost in practice.
>
> **R2:** Thank you for the problem. Similar to existing GNN [4] [5] and VI [6] [7] works, it may be not so common to measure the time complexity of these modules, as well as our GraphDE. To resolve your concern of GraphDE's efficiency, we provide the practical time cost as follows:
>
> | Backbone (GAT) | DropEdge | GraphDE-v | GraphDE-a | GRAND   |
> | -------------- | -------- | --------- | --------- | ------- |
> | 0.2114s        | 0.5701s  | 2.0962s   | 2.6869s   | 6.0335s |
>
> The table reports the training time per epoch on DrugOOD. More specifically, the maximum training epoch is set as 400, and the models usually converge and early stop at around 150 epochs. So it usually takes at around 5min to train our GraphDE model, which can achieve a 5% test accuracy improvement over the backbone. Relatively, we believe this is a valuable time-accuracy trade-off. Besides, we find that GraphDE-v is obviously faster than GraphDE-a, this is because it utilizes simple learnable scalars during training and does not need to calculate the posterior analytically. So this can be a good choice if we have limited time resource. In comparison to other two plug-in modules, we find that GraphDE is much faster than GRAND. Besides, it is slower than DropEdge but with much better testing performance.
>
> **Q3:** The experiments are not entirely convincing. a) We need to compare with OOD baselines such as IRM, DRO, and DIR. b) We need to consider the more popular and powerful backbones such as GIN. c) We need to conduct futher experiments on the OGB datasets.
>
> **R3:** a) Thank you for proposing new baselines to further complete our work. However, these papers focus on the topic of OOD generalization, which is orthogonal to our work, as we have discussed in Appendix B. Specifically, OOD generalization aims at training a model that can generalize to the unknown testing distribution from the limited training data. However, debiased learning wishes to identify the outliers (harmful examples) in the training dataset and mitigate their bad effects during training. That's why we do not adopt these baselines in our paper.
>
> b) Thank you for proposing GIN as another backbone to test our GraphDE. We acknowledge that GIN is a more powerful baseline than GCN and GAT (also shows in the experiment results) and is widely adopted in the literature. So we add new debiased learning experiment results for GIN on the DrugOOD dataset as shown in the following table. The debiased setting is the same as in the original paper (e.g. we use two layers with hidden size 64). As shown in the table, GraphDE can consistenly improve the test accuracy of GIN across all the biased ratio. In particular, it can have an improvement at around 4% at biased ratio 0.3. These results can strongly support the debiasing power of GraphDE.

---

> > ### Author Response · Authors · 2022-08-02
> > **Response to reviewer iPSD (cont.)**
> >
> > | Model     | 0              | 0.02           | 0.05           | 0.1            | 0.15           | 0.2            | 0.25           | 0.3            |
> > | --------- | -------------- | -------------- | -------------- | -------------- | -------------- | -------------- | -------------- | -------------- |
> > | GIN       | 76.78±2.87     | 75.53±1.75     | 72.20±6.66     | 73.26±4.26     | 73.14±2.84     | 73.97±2.43     | 72.60±1.99     | 71.91±5.25     |
> > | GraphDE-v | 77.00±2.91     | 76.92±1.16     | 74.78±2.70     | 75.47±2.67     | 74.37±1.33     | 74.90±2.32     | **75.67±1.95** | **75.89±1.23** |
> > | GraphDE-a | **77.18±3.08** | **77.20±2.26** | **76.02±1.81** | **76.30±2.52** | **75.36±0.43** | **76.45±1.00** | 75.60±1.07     | 75.73±2.17     |
> >
> > c) Thank you for pointing out the new testbed for our GraphDE. In fact, we have tried GraphDE on `ogbg-molhiv` (it is splitted according to different scaffolds) beforehand. Our conception is to use molecules from different scaffolds to serve as ID and OOD data. However, the train, valid, and test datasets provided by OGB are mixed with molecules from different scaffolds and cannot be captured by one distribution. This may be conflict with our assumption of ID and OOD data, making the dataset unsuitable for our experiments (and we get unsatisfying results as expected). There may exist certain interface that we can curate molecules from different scaffolds to re-conduct the experiment (our DrugOOD dataset is just another molecule dataset providing such an interface), but we leave it for future work due to the limited time of rebuttal.
> >
> > **Q4:** The writing of the paper could be improved. a) Prop 4.1 and Prop 4.2 should be defined in mathematical forms. b) The claim that modules are optimized in a mutually-promoting manner is not verified in experiments. c) The problem formulation in Section 2 seems to be literally identical to the existing work EERM.
> >
> > **R4:** a) Thank you for pointing out this problem of rigorousness. We have re-formulate the two propositions as follows:
> >
> > **Proposition 1.** 1) The learning objective for GraphDE is in a re-weighted form when $q_\phi(\mathbf e|\mathbf A,\mathbf X,\mathbf y)$ is instantiated as a Bernoulli distribution, with $q_\phi(\mathbf e_i=1|A_i,X_i,y_i)$ acting as a weight for the $i$-th sample; 2) Given the ideal recognition model $q_\phi^*$ that gives $q_\phi^*(\mathbf e=1|(A, X, y)\in\mathcal D_{in})=1$ and $q_\phi^*(\mathbf e=1|(A, X, y)\in\mathcal D_{out})=0$, the generative models can learn to best fit the ID data.
> >
> > **Proposition 2.** 1) Assuming the generative models fit to the ID data, i.e. $p_\theta(A|X\in\mathcal D_{in})\geq p_\theta(A|X\in\mathcal D_{out})$ and $p_\theta(y|(A, X)\in\mathcal D_{in})\geq p_\theta(y|(A, X)\in\mathcal D_{out})$, the recognition model will learn to predict $q_\phi(\mathbf e=1|(A,X,y)\in\mathcal D_{in})\geq q_\phi(\mathbf e=1|(A,X,y)\in\mathcal D_{out})$; 2) Given optimal generative models that best fit the ID data and perform randomly on outliers, there exists a recognition model $q_\phi^*$ which yields the minimal objective while ideally predict the environment variable.
> >
> > The corresponding propositions in the paper have also been revised in our newly updated version. Propositipn 1.1 is left unchanged as we believe the terms "re-weighted"  and "weight" are more friendly to readers to understand the rationale of GraphDE. For Proposition 2.2, we choose to state it directly in language, since the mathematical definitions are clear from the context.
> >
> > b) Sorry for causing the ambiguity. In fact, these two claims have both been discussed in our main text in Section 4.3. **For mutually-promotion, you can refer to Figure 4.(b) and Figure 4.(c)**. Specifically, we study the debiasing performance with/without structure estimation module in Figure 4.(b) and find that the test accuracy drops over 6% at biased ratio 0.25. In Figure 4.(c), we study the detection performance with/without the classification module and find that the AUROC drops approximately 3% at biased ratio 0.3. These two results support our claim that the composed modules in GraphDE are optimized in a mutually-promoting manner. **For the inferred environments, you can refer to Figure 5.(b)**. The caption "VI probability" on the x-axis denotes $q_\phi(\mathbf e=1|(A,X,y)\in\mathcal D^{tr})$ and we have visualized the distribution of the inferred environment variable on SPMotif and DrugOOD. Typically, we can see that the OOD data are generally assigned with a lower probability score than the ID data, which suggests that GraphDE has taken effect during the training process. You can learn more details referring to our discussions in Section 4.3.
> >
> > c) We are really sorry for the oversight. We have revised our presentation in the newly updated paper to compensate for this issue.

---

> > > ### Author Response · Authors · 2022-08-02
> > > **Response to reviewer iPSD (cont.)**
> > >
> > > **Q5:** Missing related works on graph debiased learning.
> > >
> > > **R5:** Thank you for pointing out the missing related works for our paper. In fact, we have surveyed these two papers beforehand. **The important term "bias" does appear in these two papers, but has entirely different meaning to our paper**. Specifically, they are studying about **the training sample selection bias**. In SR-GNN [1], the authors claim that the bias in the sampling process to select nodes for training can create distributional differences between training and testing set, and they use a regularization and an instance reweighting component to address this issue. In DGNN [2], the authors focus on the label selection bias. Specifically, their datasets are created by biased select nodes from different classes, or select equal but small number of nodes from each class. And they resolve this issue by their proposed differentiated decorrelation regularizer in a causal view. We also focus on the graph-level but not node-level classification tasks in [1] [2]. In this paper, however, the "bias" is more correlated to the meaning in [3]. Specifically, as we discussed in Section 1, **our "bias" denotes the outliers (harmful samples)** that will skew the training process in the training data. Therefore, our debiased learning is to identify the outliers and mitigate their effects  to promote the classification accuracy on ID data, different from OOD generalization in these two papers.
> > >
> > > Also thank you for pointing out the minor problems and we will fix the issues upon revision/publication.
> > >
> > >
> > > ### References
> > > [1] Shift-Robust GNNs: Overcoming the Limitations of Localized Graph Training Data, NeurIPS 2021.
> > >
> > > [2] Debiased Graph Neural Networks with Agnostic Label Selection Bias, TNNLS 2022.
> > >
> > > [3] Resolving Training Biases via Influence-based Data Relabeling, ICLR 2022.
> > >
> > > [4] Semi-Supervised Classification with Graph Convolutional Networks, ICLR 2017.
> > >
> > > [5] Graph Attention Networks, ICLR 2018.
> > >
> > > [6] Robust Variational Autoencoders for Outlier Detection and Repair of Mixed-Type Data, AISTATS 2020.
> > >
> > > [7] A Flexible Generative Framework for Graph-based Semi-supervised Learning, NeurIPS 2019.

---

> ### Author Response · Authors · 2022-08-09
> **To Reviewer iPSD: Looking foward to your reply**
>
> While the other reviewers have acknowledged our rebuttal and raised their rating accordingly, we are wondering whether our responses have addressed your concerns correctly. Your feedback would be really appreciated and will definitely help reach an informative decision on our submission. Thank you!

---

### Official Review · Reviewer_bxvk · 2022-07-11

**Rating:** 6
**Confidence:** 3
**Soundness:** 3 good
**Presentation:** 3 good
**Contribution:** 3 good

**Summary:**

In this work the authors propose a method for jointly addressing outliers present in training data as well as learning an OOD detection model. Using a simple binary environment variable they learn a unified probability mixture model and demonstrate performance on both the debiased learning and OOD detection tasks.

**Questions:**

1. In line 42, do you actually agree with the statement that test time is just about feature/covariate shift? I would argue the very premise of a need for a method such as yours, plus basic knowledge of real world data characteristics together suggest that joint distribution shift is also a test time issue (I know it's arguably hard to address directly with any method)
2. For the generative structural process you are making the assumption that the connectivity, or the adjacency matrix, is implied (caused) by the node features - cites random graph theory/homophily. Is this a valid assumption? Under this assumption it seems to follow that the distribution shift is therefore caused by node feature shift. Could your method be extended to when this assumption does not hold?
3. Figure 3 and Table 1 seem to display the same kind of data? Could they be visualized the same way, or was this choice made because the bias ratio was fixed for the table? (why was this fixed for that table?)

**Limitations:**

1. This method introduces the requirement for a choice of prior on the cleanliness of the training set - this will have some effect on the final model performance, unknown in a real deployment setting.


**Strengths And Weaknesses:**

### Strengths

**Quality**: The joint perspective on training time outliers and the task of OOD detection at test time is principled and correct, in this reviewer's opinion. The choice of datasets and the analyses are appropriate for the problem setup and the augmentation methods are relevant baselines. The explanatory ablation results at the end help address appropriate questions about the design choices made.

**Originality**: The main novelty comes from the end-to-end manner in which they address weighting examples at training time as well as extracting the OOD model

### Weaknesses

**Clarity**: The section where Propositions 1 and 2 are stated is lacking proper backing for its claims, or rather in its current form, it's unclear whether this section contributes meaningfully. Overall, the concept of predicting the binary environment variable in order to learn a more optimal model conditioned on the environment variable makes sense, but the assumptions on the perfect recognition and perfect generative models are too strong to just state in this section. I don't feel the sections add significant justification for the approach.

**Significance**: Results on more updated large scale data (say derived from OGB) would help demonstrate that the binary environment variable approach doesn't break down at scale - which matters for practical adoption of new methods, especially those focused on real world problems like distribution shift.

Work would generally benefit from a close editing pass from a native english speaker, _but this does not factor into my assessment_.

---

> ### Author Response · Authors · 2022-08-02
> **Response to reviewer bxvk.**
>
> Thank you for your time and valuable advice. We are glad that you acknowledged our insight, experiments, and novelty. Here are our responses to your questions:
>
> **Q1:** Results on more updated large scaled data (OGB) would help demonstrate that the binary environment variable approach doesn't break down at scale.
>
> **R1:** Thank you for pointing out the new testbed for our GraphDE. In fact, we have tried GraphDE on `ogbg-molhiv` (it is splitted according to different scaffolds) beforehand. Our conception is to use molecules from different scaffolds to serve as ID and OOD data. However, the train, valid, and test datasets provided by OGB are mixed with molecules from different scaffolds and cannot be captured by one distribution. This may be conflict with our assumption of ID and OOD data, making the dataset unsuitable for our experiments (and we get unsatisfying results as expected). There may exist certain interface that we can curate molecules from different scaffolds to re-conduct the experiment (our DrugOOD dataset is just another molecule dataset providing such an interface), but we leave it for future work due to the limited time of rebuttal. Besides, it is worth noting that we are focusing on graph-level classification tasks. These tasks generally deal with a set of graphs with a relatively small number of nodes (10~1k), and will not encounter the scalability issue (which is common for node-level tasks) since it can train the model through mini-batch optimization.
>
> **Q2:** In line 42, do you actually agree with the statement that test time is just about feature/covariate shift?
>
> **R2:** Sorry for causing the ambiguity. Typically, we agree  that joint distribution shift is also a test time issue, and GraphDE also models the joint distribution of $(\mathbf A,\mathbf X,\mathbf y, \mathbf e)$ for the testing data as in Eq. (8) and Eq. (9). In line 42, however, we mention that "**OOD testing samples are solely determined by their features**" under the semantics of OOD detection. That is, we know about both the features and labels for the training data, and the **training outliers can be derived from abnormal features or flipping labels**. But for the testing data, since we have no idea of their ground truth labels, we need to **determine whether they are OOD or not just based on their features when conducting OOD detection** [1] [2]. It is a difference between training and testing data. As shown in Eq. (8), we compute $p_\theta(\mathbf e|\mathbf A,\mathbf X)$ for testing data, conditional on the graph features $\mathbf A$ and $\mathbf X$.
>
> **Q3:**  Is it a valid assumption that the connectivity, or the adjacency matrix, is implied by the node features? Under this assumption it seems to follow that the distribution shift is therefore caused by node feature shift. Could GraphDE be extended to when this assumption does not hold?
>
> **R3:** In fact, we can also change the assumption to model $p(\mathbf X|\mathbf A)$ using techniques such as Dirichlet energy [3], which will give a foundamentally different model from GraphDE. However, we believe the assumption that the connectivity is implied by the node features is enough expressive and general from the perspective of graph generation, which is widely adopted in the graph learning community [4] [5]. Besides, modeling $p(\mathbf A|\mathbf X)$ **does not necessarily mean that the distribution shift is caused by node feature shift**. Since the conditional distribution is a function of both $\mathbf A$ and $\mathbf X$, it can capture the distribution shift on the adjcency matrices as well. For cases when this assumption does not hold, GraphDE can also take effect since it only requires a dependence of $\mathbf A$ on $\mathbf X$, not necessarily a functional relationship.
>
> **Q4:** Figure 3 and Table 1 seem to display the same kind of data? Could they be visualized in the same way, or was this choice made because the bias ratio was fixed for the table? (why was this fixed for that table?)
>
> **R4:** Figure 3 and Table 1 show experiment results on different datasets, while giving out information of different dimensions. Note that DropEdge, Grand, GraphDE-v, and GraphDE-a are all plug-in modules for the GNNs. In Figure 3, we focus on **the test performance degradation w.r.t. different biased ratio**, and all the plug-in modules are applied on the GAT backbone for fair comparison. In Table 1, however, we apply the plug-in modules to all the four GNN backbones. We fix the biased ratio in this case to show that **GraphDE outperforms the other plug-in modules on top of all the GNN backbones**. To summarize, we use Figure 3 to show that GraphDE can consistently outperform the baselines on different biased ratios, and use Table 1 to prove GraphDE takes effects across all the GNN backbones.

---

> > ### Author Response · Authors · 2022-08-02
> > **Response to reviewer bxvk.**
> >
> > ### References
> >
> > [1] Deep Anomaly Detection with Outlier Exposure, ICLR 2019.
> >
> > [2] Energy-based Out-of-distribution Detection, NeurIPS 2020.
> >
> > [3] Laplacian Eigenmaps and Spectral Techniques for Embedding and Clustering, NeurIPS 2002.
> >
> > [4] A Flexible Generative Framework for Graph-based Semi-supervised Learning, NeurIPS 2019.
> >
> > [5] Graph Stochastic Neural Networks for Semi-Supervised Learning, NeurIPS 2020.

---

> ### Comment · Reviewer_bxvk · 2022-08-09
> **Acknowledgement of author rebuttal**
>
> Thank you to the authors for their detailed response.
>
> Regarding the question on scaling to larger datasets, I still believe this is a potential issue with the method, but understand the authors argument about accessibility of properly split data and also the still-standard practice of focusing on small sizes for graph classification. The clarification for Q2 is appreciated, but of course I'd prefer that this part was stated more carefully in the intro - this is the author's prerogative however.
>
> Responses to Q3/4 resolved ambiguities for me.
>
> I'm happy to move this from borderline to a weak accept, 6.

---

> > ### Author Response · Authors · 2022-08-09
> > **Thank you**
> >
> > Thank you for the positive feedbacks. We have polished our presentation according to your advice in the revised version. We believe these updates can help the readers understand our work better.

---

### Official Review · Reviewer_gt4u · 2022-07-13

**Rating:** 7
**Confidence:** 3
**Soundness:** 4 excellent
**Presentation:** 3 good
**Contribution:** 3 good

**Summary:**

This paper focuses on the problem of debiasing learning and out-of-distribution detection in graph data. The authors argue that existing works typically address these two tasks independently, but the intrinsic connections between training outliers and OOD test samples are overlooked. To this end, the authors propose a novel model called GraphDE to tackle debiasing learning for training data and OOD detection for test data under a unified probabilistic model. Extensive experiments are conducted on different GNN backbones, and the results validate the superiority of the proposed method over the baselines. Theoretical analyses are also provided to justify the effectiveness of the model.

**Questions:**

Why does not GraphDE outperform OCGIN or other baselines on Collab in terms of AUPR and on DrugOOD in terms of FPR95? More discussions on the undesirable results may be helpful.

**Limitations:**

No potential negative societal impact.

**Strengths And Weaknesses:**

Strengths:
1. This paper focuses on an interesting and important problem. Outliers in training data and OOD samples are very common in the graph domain. Properly handling them is important to ensure the performance of a GNN model in practical usage.

2. The paper manages to solve the problem of graph debiased learning and OOD detection in a novel perspective. That is, a unified framework is proposed to model the generative process of both the training data and the test data, where the two problems are tackled dependently.

3. The paper is well-written and easy for readers to follow.

4. The experimental results show that the proposed method achieves consistent performance improvements over the baselines, which demonstrates its superiority. Also, a detailed theoretical analysis is provided to justify the rationale of GraphDE.

Weaknesses:
1. It would be better if the authors can discuss under what condition the proposed model may not achieve desirable performance. For example, why GraphDE cannot achieve the best performance on Collab in terms of AUPR and on DrugOOD in terms of FPR95?

---

> ### Author Response · Authors · 2022-08-02
> **Response to reviewer gt4u.**
>
> Thank you for the comments and nice suggestions. We are pleased that you are satisfied with our motivation, methodology, theory, writing, and experiments. Here is our response to your question:
>
> **Q1:** It would be better if the authors can discuss under what condition the proposed model may not achieve desirable performance.
>
> **R1:** It can be found from Table 2 that GraphDE beats all the other baselines on 7 out of 9 benchmarks. Specifically, GraphDE does not achieve the best performance on Collab in terms of AUPR (area under the precision-recall curve) and on DrugOOD in terms of FPR95 (false positive rate when the true positive rate is 95%). The reason for this can be subtle associated with the detector, dataset, and the metric.
>
> 1) For Collab, since we divide ID/OOD data according to graph sizes, there may be graphs of similar sizes in the testing ID/OOD datasets. Besides, the graph sizes can also vary in the ID/OOD dataset. The result probability score distribution is shown in the middle of Figure 6, both ID/OOD distribution have two peaks and overlap to some extent. This may decrease the area under the precision-recall curve since there is not a point to separate out ID/OOD data ideally.
> 2) For DrugOOD, the ID/OOD data is decided based on molecule scaffolds. This measure is somewhat unclear and molecules from different scaffolds can also have common properties. The result probability score distribution is shown in the right of Figure 6, in which ID and OOD distributions are very close to each other. In this sense, the false positive rate may be high even when we have a high true positive rate (at 95%).

---

### Author Response · Authors · 2022-08-02
**General Response**

Dear Area Chair and Reviewers,

We appreciate reviewers' precious time and valuable advice. We are happy that most of reviewers acknowledged our motivation (gt4u, bxvk, iPSD, xAwj, hmWz), writing (gt4u, bxvk, hmWz), novelty (gt4u, bxvk, xAwj, hmWz) and experiments (gt4u, bxvk, xAwj, hmWz). The major concerns lie in our novelty (iPSD) and additional experiments asked by bxvk, iPSD, and xAwj. To clarify some potential misunderstandings of our paper, we first address some shared concerns:

- **Novelty.** VI and GNNs are two fields that have been deeply studied in the literature. However, we emphasize that this paper's main novelty does not lie in these two conventional methodologies. As we have claimed in Section 1, in this paper, we focus on the debiasing and OOD detection problems for GNNs: 1) Most importantly, we model the graph generative process and therefore propose a unified probabilistic framework to define and tackle these two problems simultaneously. 2) Besides, through introducing a two-component structure for the generative models, we induce a novel learning objective and justify that GraphDE can identify and down-weight outliers during training while provide an OOD detector on the test set.
- **Experiment baselines.** It is worth noting that previous works such as IRM, DRO, DIR, and OOD-GNN focus on the topic of OOD generalization, which is fundamentally different from the debiasing learning we study in the paper (which we have discussed in Appendix B). Specifically, OOD generalization aims at training a model that can generalize to the unknown testing distribution from the limited training data. However, debiased learning wishes to identify the outliers (harmful examples) in the training dataset and mitigate their bad effects during training. So these works are orthogonal to our paper and we do not adopt them as our baselines.

We provide extraordinary experiment results and detailed answers to all the questions raised by the reviewers in the following individual responses. Besides, we have also revised the paper w.r.t. the suggestions of the reviewers, which are highlighted in blue in the newly submitted version.

---

### Author Response · Authors · 2022-08-06
**Inquiry for post-rebuttal comments**

Dear reviewers,

I would like to express our sincere gratitude for your constructive advice on this paper.

Since the discussion period is approaching its ending, we would be glad to hear from you about whether our rebuttal has addressed your concerns? If you have any further questions and concerns, feel free to post your comments so that we can respond to your questions and concerns.

We will really appreciate it if you could post some comments so that we can improve this paper accordingly.

---

### Meta-Review · Area_Chair_GwsJ · 2022-08-30

**Recommendation:** Accept
**Confidence:** Less certain

**Metareview:**

The authors propose a mixture modeling approach to train GNNs so that out-of-distribution data can be properly down-weighted during training and detected during testing. The reviews were mixed, with some reviewers criticizing the technical novelty and experimental comparison. Indeed, the authors could have explained their contribution more transparently, and emphasized a bit more on the new challenges in the GNN setting, which the response has largely addressed. Perhaps it is also worthwhile to discuss classic works on mixture of experts, as well as variational Bayesian approaches (e.g. https://ieeexplore.ieee.org/document/5563102). As to the experimental comparison, I think the authors made some good explanations in the response and it is perhaps too ambitious for anyone to compare to every possible alternative.

In the end, we think the application of the mixture modeling approach to GNNs is sufficiently interesting, and the initial experimental results appear to be encouraging. We urge the authors to further revise their work by incorporating all changes during the response and better positioning the contributions in historical context.

**Award:**

No

---

### Decision · Program_Chairs · 2022-09-14

Accept